# Mechanical sensitivity of Piezo1 ion channels can be tuned by cellular membrane tension

Amanda H Lewis, Jörg Grandl*

Department of Neurobiology, Duke University Medical Center, Durham, United States

**Abstract** Piezo1 ion channels mediate the conversion of mechanical forces into electrical signals and are critical for responsiveness to touch in metazoans. The apparent mechanical sensitivity of Piezo1 varies substantially across cellular environments, stimulating methods and protocols, raising the fundamental questions of what precise physical stimulus activates the channel and how its stimulus sensitivity is regulated. Here, we measured Piezo1 currents evoked by membrane stretch in three patch configurations, while simultaneously visualizing and measuring membrane geometry. Building on this approach, we developed protocols to minimize resting membrane curvature and tension prior to probing Piezo1 activity. We find that Piezo1 responds to lateral membrane tension with exquisite sensitivity as compared to other mechanically activated channels and that resting tension can drive channel inactivation, thereby tuning overall mechanical sensitivity of Piezo1. Our results explain how Piezo1 can function efficiently and with adaptable sensitivity as a sensor of mechanical stimulation in diverse cellular contexts.

## Introduction

Mechanosensation is essential for cells ranging from bacteria, which must regulate cell volume in response to harsh osmotic stress, to Merkel cells and sensory neurons in human fingertips, which are exquisitely sensitive to gentle touch (*Ranade et al., 2015*). Mechanosensation is initiated through the opening of mechanosensitive ion channels, including the $K_{2P}$ family in vertebrates, NOMPC in *Drosophila*, and the DEG/ENaC family in *Caenorhabditis elegans* (*Ranade et al., 2015*). Recently, Piezo proteins were identified as the pore-forming subunits of an excitatory (non-selective cation) mechanosensitive channel in metazoans (*Coste et al., 2010*; *2012*; *Faucherre et al., 2013*; *Kim et al., 2012*; *Schneider et al., 2014*). Piezos are large proteins with >2500 residues that lack homology to any known proteins (*Coste et al., 2015*). The two mammalian isoforms, Piezo1 and Piezo2, are widely expressed and play key roles in many physiological processes, including vascular development, red blood cell volume regulation, lineage choice in neural stem cells, and touch sensation in Merkel cells and DRG neurons (*Cahalan et al., 2015*; *Li et al., 2014*; *Maksimovic et al., 2014*; *Pathak et al., 2014*; *Ranade et al., 2014a*; *Woo et al., 2014*). In mouse, knockout of either isoform is lethal, further emphasizing the functional importance of the protein (*Ranade et al., 2014a*).

Despite an increased understanding of the various roles Piezos play in many biological processes, the activation mechanism, including the precise physical stimulus that initiates pore opening (gating), is unknown. The recent medium-resolution cryo-electron microscopy structure of mouse Piezo1 revealed many features of the coarse channel architecture, but does not provide conclusive clues about the activation mechanism (*Ge et al., 2015*). In vivo, Piezos respond to diverse forces, including laminar flow and cellular compression (*Lee et al., 2014*; *Li et al., 2014*; *Ranade et al., 2014a*). In

*For correspondence: grandl@neuro.duke.edu

**Competing interests:** The authors declare that no competing interests exist.

**eLife digest** Piezo ion channels are proteins that are embedded in the cell membranes of many types of tissue, including the heart, lung, skin and kidney. These proteins are essential for many biological processes, including sensing gentle touches and ensuring that blood vessels develop properly.

When stimulated by mechanical forces, a central pore in the Piezo channel opens to allow positively charged ions to flow into the cell, which triggers electrical and chemical signaling processes inside the cell. However, it was not known exactly what type of mechanical stimulus is sensed by Piezo ion channels.

Lewis and Grandl expressed Piezo ion channels in cultured human kidney cells, and opened them by applying pressure to parts of the cell membrane inside a glass pipette. This causes a number of changes to the membrane, including to its curvature and tension, either of which could potentially open the Piezo channels. However, Lewis and Grandl were able to calculate from images of the cell membrane inside the pipette that tension is the activating stimulus.

Further experiments unexpectedly revealed that the tension that is usually present in the cell membrane is sufficient to inactivate Piezo channels and prevent them from responding to an additional mechanical stimulus. This suggests that Piezo ion channels are inherently more sensitive to tension than previously realized, which could explain why different cell types appear to have different sensitivities to pressure.

Although Lewis and Grandl have now shown that Piezo channels are activated by tension, more work is needed to investigate how the Piezo ion channel senses this force, and how this leads to the channel pore opening.

heterologous systems, two techniques used to evoke channel activity are direct stimulation of the cell by touching with a blunt glass pipette and application of negative pressure to stretch the membrane in a patch pipette (*Coste et al., 2010*). Both stimuli induce several geometric and energetic changes in the membrane, including alterations in curvature and tension, any of which could in principle activate mechanically activated ion channels (*Sukharev and Corey, 2004*). For example, lateral membrane tension is the stimulus for the well-characterized bacterial mechanosensitive ion channel MscL (*Moe and Blount, 2005*; *Sukharev et al., 1999*). To date, no accessory proteins of Piezos have been identified that could tether the channel to the cellular matrix, suggesting that, as for MscL, the activating stimulus may be directly transmitted through the bilayer.

A key feature of Piezos is that during a sustained stimulus, currents decay (inactivate) with a typical time course of tens of milliseconds, suggesting continuous modulation of channel availability (*Coste et al., 2010*). Consistent with a high physiological importance for inactivation, Piezo mutations that alter inactivation kinetics are linked to several human diseases, including dehydrated hereditary stomatocytosis, xerocytosis, Marden-Walker and Gordon syndromes, and distal arthrogryposis (*Albuisson et al., 2013*; *Bae et al., 2013*; *Coste et al., 2013*; *McMillin et al., 2014*). Importantly, several of these gain-of-function mutations not only reduce the rate and/or extent of inactivation, but also apparently sensitize Piezos to pressure (*Bae et al., 2013*). Additionally, the only known agonist for Piezo ion channels, Yoda1, both sensitizes Piezo1 to pressure and slows inactivation (*Syeda et al., 2015*).

The pressure sensitivity of Piezos also varies with cell type, with previous reports of half-maximal pressure for activation for Piezo1 ranging from $\sim-15$ to $-40$ mmHg (*Coste et al., 2010*; *Li et al., 2014*; *Pathak et al., 2014*). Local membrane tension and curvature also vary among and even within cells, with potentially important implications for Piezo function. Piezo sensitivity is also modulated by proteins including EPAC1 and STOML3; differential expression of these and other modulators both among cells and within a single cell may also regulate the overall sensitivity (*Eijkelkamp et al., 2013*; *Poole et al., 2014*). A mechanistic understanding of the link between the activating stimulus, local membrane environment, inactivation, and channel sensitivity may provide specific strategies for pharmacological modulation of Piezo activity. Here, we combine high-resolution, high-contrast imaging with electrophysiology to investigate whether membrane curvature or lateral membrane tension is the physical stimulus for activation of Piezo1 and how sensitivity to this stimulus might be

intrinsically regulated. We find that Piezo1 is activated by membrane tension with a high degree of sensitivity ($T_{50}$ = 1.4 ± 0.1 mN/m) and that this sensitivity is directly influenced by resting membrane tension.

## Results

### Piezo1 is activated by both convex and concave membrane curvature in multiple patch configurations

In order to assay the influence of membrane curvature and lateral membrane tension on Piezo ion channel activity, we transiently transfected HEK293t cells with mouse Piezo1-IRES-GFP and performed electrophysiological recordings during application of negative or positive pressure to the membrane patch by using a high-speed pressure clamp system, while simultaneously imaging the membrane inside the patch pipette with high resolution (400x), differential interference contrast (DIC) microscopy.

Consistent with previous reports, we observed that in cell-attached patches, negative pressure induced rapidly-inactivating inward currents at −80 mV that are characteristic of Piezo1 ion channels (*Coste et al., 2010*; *2012*). Currents first became apparent at −5 mmHg, increased with the magnitude of pressure, and reached saturation at ~−30 mmHg (*Figure 1A*). As expected, our simultaneous imaging approach revealed that negative pressure also induced a convex curvature of the membrane and that the radius of curvature decreased with increasing magnitudes of pressure (*Figure 1A*; *Video 1*).

Next, we wanted to probe how Piezo1 would respond to opposite (concave) membrane curvature. We therefore applied positive pressure to our membrane patches, which indeed inverted the membrane curvature. Piezo1 currents were also reliably induced by this positive pressure protocol (*Figure 1B*). Peak current amplitudes again increased with the magnitude of pressure, albeit with a striking difference: pressure responses were right-shifted, as currents first became apparent at +15 mmHg and often did not reach saturation before rupture of the patch. Interestingly, we also often observed small currents upon release of small (+5 mmHg) positive pressure stimuli, discussed further below. Together, these observations suggest that in cellular membranes, Piezo1 is activated by both convex and concave membrane curvature.

We next wanted to test if membrane sidedness and mechanical stability conferred by the cytoskeleton had any influence in this activation process. We therefore repeated these experiments in both inside-out and outside-out patches, both of which result in disrupted cytoskeletal structure and in the latter, inverted membrane leaflets with respect to the patch pipette. For inside-out patches, both positive and negative pressure again evoked transient inward currents through Piezo1 ion channels (*Figure 1C and D*). In this patch configuration, the optical density of the membrane was reduced compared to cell-attached patches, consistent with the idea that less cytoskeleton is retained in a cell-detached configuration (*Suchyna et al., 2009*). However, DIC imaging was still sufficiently sensitive to resolve the convex and concave membrane curvatures induced by positive and negative pressure protocols, respectively. In the outside-out configuration, we were only able to consistently visualize the membrane during application of positive pressure, but not during negative pressure, perhaps because excess membrane folded within the tip was conformationally flexible and therefore not resolved at our imaging speed (*Ruknudin et al., 1991*). Still, we obtained the same result as for the other patch configurations: both positive and negative pressure robustly activated Piezo1 currents (*Figure 1E and F*). In contrast to cell-attached and inside-out patches, we did not consistently observe decay of currents elicited in the outside-out configuration.

In all of the above experiments, only negligible currents were elicited in cells transfected with empty vector (pcDNA), showing that the currents induced by membrane curvature are specifically mediated by Piezo1 (*Figure 2A–C*). While Piezo1-mediated currents were reliably evoked by bidirectional pressure in all patch configurations, individual patches did show some expected variability in their precise current amplitude levels, mostly among different patch configurations. Several variables could potentially change among configurations, including patch surface area, cytoskeletal content, and stability of the gigaseal, any or all of which could in theory contribute to the observed differences. Specifically, currents elicited by positive pressure in outside-out patches were typically larger than in the other configurations, consistent with the observation that a larger surface area of

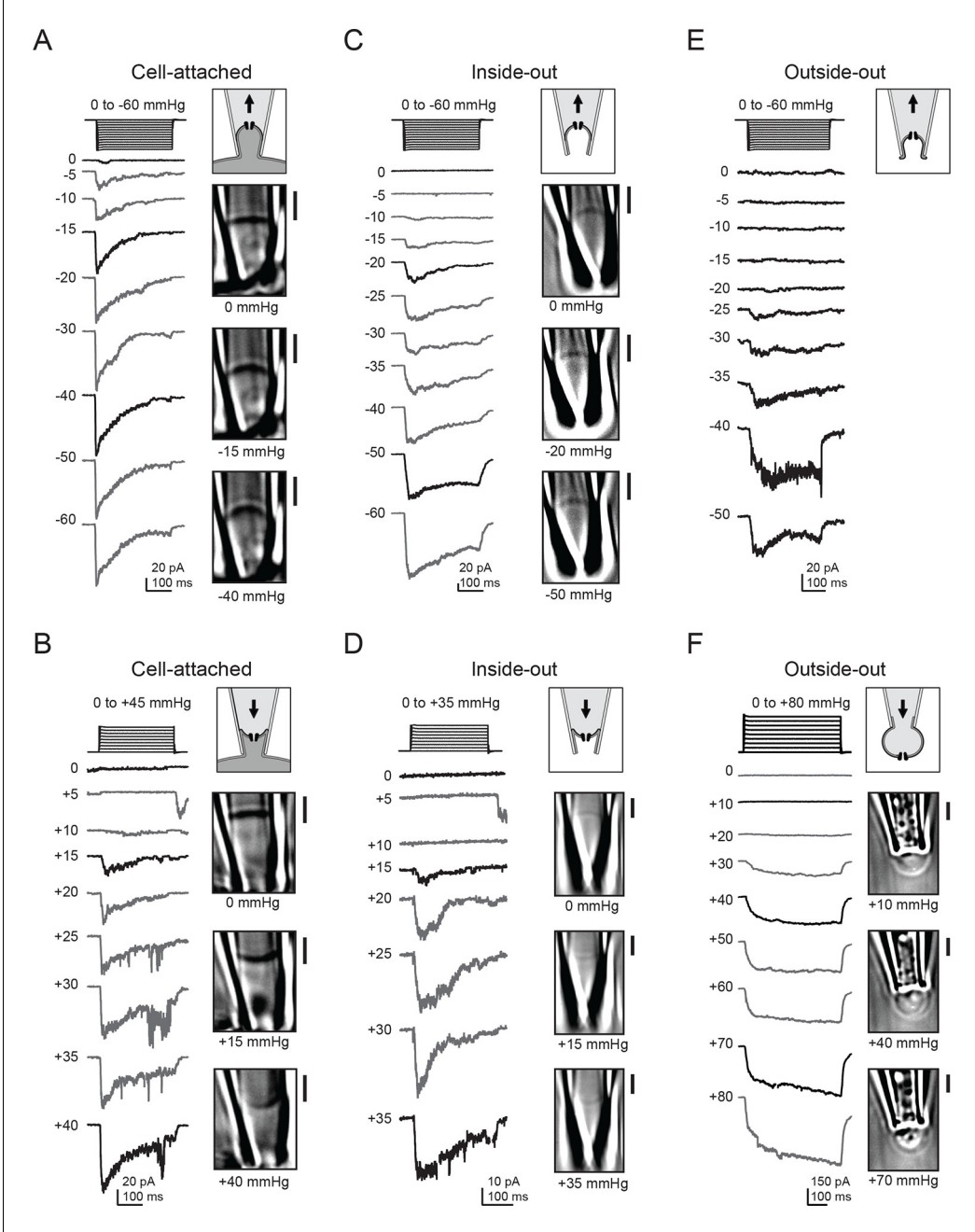

**Figure 1.** Electrophysiology and high-contrast imaging of Piezo-containing membranes. Pressure-step protocol, representative currents and corresponding images from individual cell-attached patches from a HEK293t cell expressing mouse Piezo1-IRES-GFP, upon negative (**A**) and positive (**B**) pressure stimulation. Pressure-step protocol, respective representative currents and corresponding images from individual inside-out patches upon negative (**C**) and positive (**D**) pressure stimulation. Pressure-step protocol, respective representative currents and images from individual outside-out patches upon negative (**E**) and positive (**F**) pressure stimulation. All patches were held at −80 mV. Scale bars are 2 μm for all images.

membrane is preserved within the confines of the gigaseal in this configuration (*Figure 1F*). In contrast, currents elicited by positive pressure in inside-out patches were small, likely because patches that survived multiple positive pressure pulses were biased towards those made using smaller pipettes (*Figure 1D*).

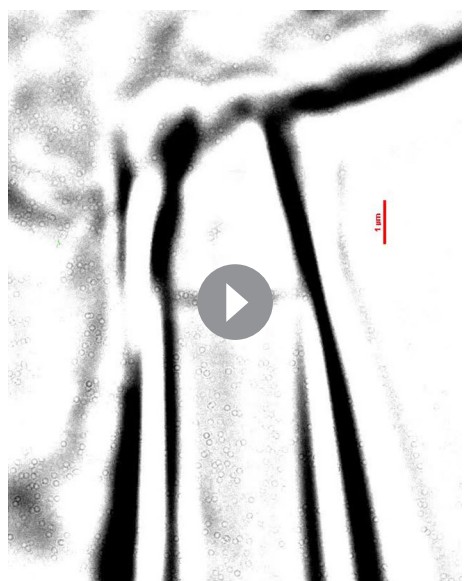

**Video 1.** Response of cell-attached HEK293t patch to stimulation with negative pressure in −5 mmHg increments; acquired at 7.9 frames/s, played at 50 frames/s. Video corresponds to cell in *Figure 1A*.

We also observed differences in the pressure required for half-maximal activation ($P_{50}$) of Piezo1 among patch configurations and directions of curvature (*Figure 2D*). Cell-attached patches stimulated with negative pressure required less pressure for activation ($P_{50}$ = −16.7 ± 2.8 mmHg; N=15) than inside-out patches ($P_{50}$ = −35.8 ± 4.0 mmHg; N=10; P<0.05). In contrast, cell-attached and inside-out patches required similar pressures for activation with positive pressure (cell-attached: $P_{50}$ = +19.3 ± 1.2 mmHg; N=12; inside-out: $P_{50}$ = +26.7 ± 4.2 mmHg; N=7; P>0.05). These values for positive pressure are likely an underestimate of the true $P_{50}$ values for the positive pressure in both configurations, due to premature rupture of some patches before reaching saturation. For outside-out patches, much larger pressures were required for activation than other configurations (positive pressure: $P_{50}$ = +70.0 ± 9.6 mmHg; N=11; negative pressure: $P_{50}$ = −39.8 ± 3.1 mmHg; N = 6). Together, these experiments provide strong evidence that Piezo1 can be activated by both convex and concave membrane curvature. Further, as cytoskeletal content varies with patch configuration, the different sensitivities we observed among configurations and with convex versus concave geometry suggests that Piezo1 sensitivity to a given stimulus (in this case, pressure) may vary significantly with the amount of cytoskeletal content and its sidedness, i.e., whether the cytoskeleton is subjected into a convex vs. concave geometry (*Suchyna et al., 2009*).

## Piezo1 activation is consistent with membrane tension as the activating stimulus

The radius of curvature (R) of a surface exposed to a pressure difference (Δp) is directly related to lateral tension (T), as described by Laplace's law: T = R·Δp/2. However, while curvature can be either positive or negative (convex or concave), tension is a symmetrical quantity. The fact that Piezo1 ion channels respond well to both convex and concave curvature therefore suggests qualitatively that Piezo1 might be activated by lateral membrane tension.

To investigate this symmetric relationship quantitatively, we next measured the radius of membrane curvature (R) for each patch and pressure step (Δp) with a custom script written in Igor-Pro (WaveMetrics, Lake Oswego, OR) and calculated the membrane tension using Laplace's law (*Figure 3A*). We focused our analysis on the cell-attached and inside-out configurations, which had the highest quality images (*Figure 1A and C*).

In order to pool data from multiple cells, we normalized each individual patch to its plateau current in response to saturating stimuli (obtained from a Boltzmann fit; see Materials and methods) and calculated current amplitude histograms as a function of membrane tension (*Figure 3B and C*). The binned data were then fit, using the standard deviations for each bin to weight the fit. For cell-attached patches, we found that Piezo1 channel activity is well described by a Boltzmann function with a tension of half-maximal activation $T_{50}$ = 2.7 ± 0.1 mN/m and a slope factor k = 0.8 ± 0.1 (N = 15). Similarly, for inside-out patches, the distribution was also well-described by a Boltzmann function; however, patches required slightly more tension for activation ($T_{50}$ = 4.7 ± 0.3 mN/m; N = 10; P<0.001 vs cell-attached) and had a shallower slope factor (k = 1.2 ± 0.1; P<0.001 vs. cell-attached).

Notably, in both configurations the likelihood of a given pulse leading to patch rupture increased sharply starting around 10 mN/m; this is consistent with the lytic tension of the gigaseal, previously reported to be ~10 mN/m (*Suchyna et al., 2009*). The fact that inside-out patches require a greater increase in tension to open Piezos is also consistent with the notion that the intrinsic resting tension

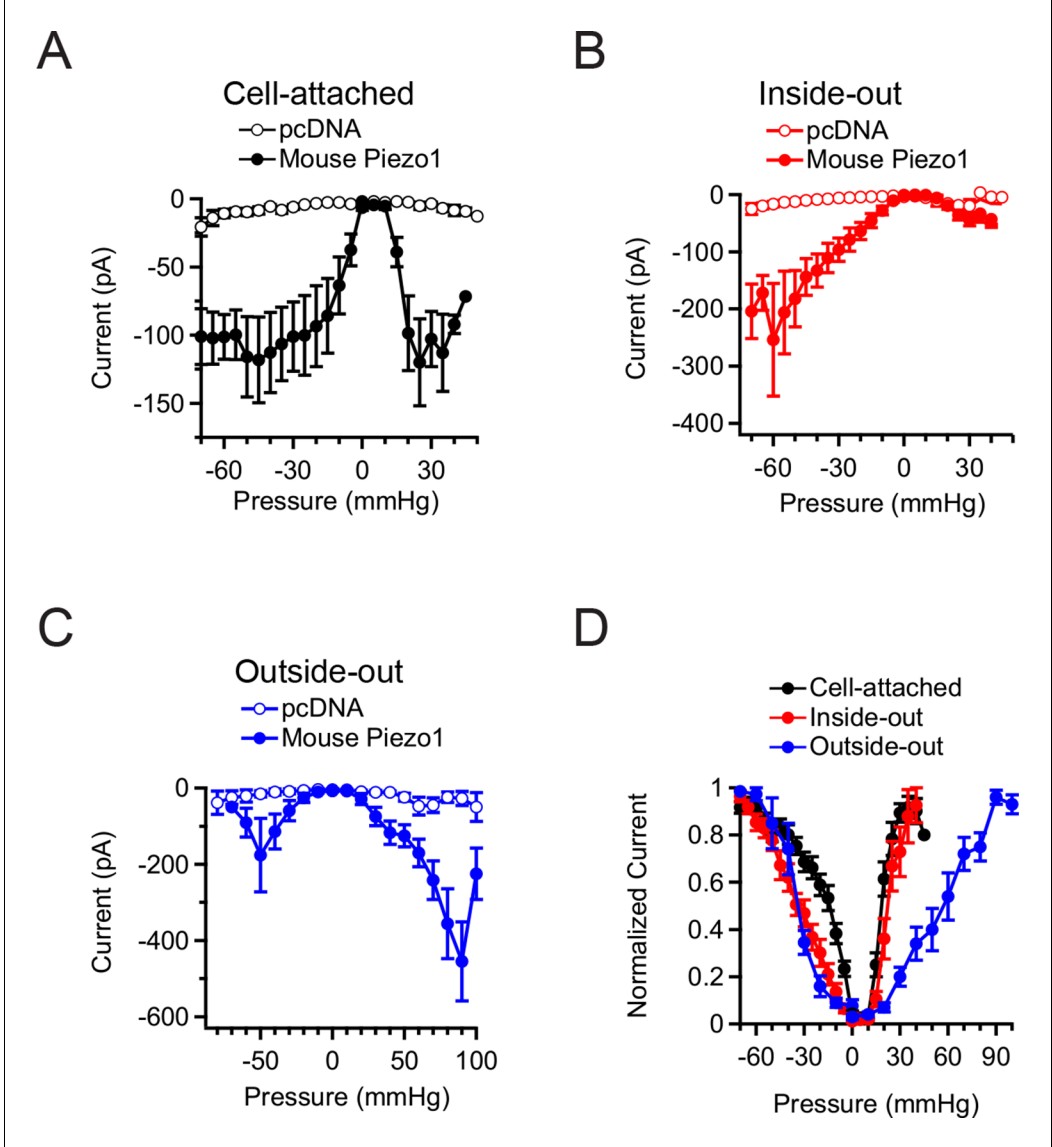

**Figure 2.** Mean Piezo1 current responses for all patch configurations upon positive and negative pressure stimulation. (**A**) Pressure-evoked currents from cell-attached patches from HEK293t cells expressing empty vector (pcDNA; open circles) or Mouse Piezo1-IRES-GFP (closed circles). Separate patches were tested for positive and negative pressure stimulation. N = 7 for pcDNA at negative pressure, N = 6 for pcDNA at positive pressure, N = 15 for Piezo1 at negative pressure and N = 12 for Piezo1 at positive pressure. (**B**) Pressure-evoked currents from inside-out patches from HEK293t cells expressing empty vector (pcDNA; open circles) or Mouse Piezo1-IRES-GFP (closed circles). Separate patches were tested for positive and negative pressure stimulation. N = 4 for pcDNA at negative pressure, N = 3 for pcDNA at positive pressure, N = 10 for Piezo1 at negative pressure and N = 7 for Piezo1 at positive pressure. (**C**) Pressure-evoked currents from outside-out patches from HEK293t cells expressing empty vector (pcDNA; open circles) or Mouse Piezo1-IRES-GFP (closed circles). Separate patches were tested for positive and negative pressure stimulation. N = 3 for pcDNA at negative pressure, N = 7 for pcDNA at positive pressure, N = 6 for Piezo1 at negative pressure and N = 11 for Piezo1 at positive pressure. (**D**) Normalized mean current-pressure relations for all six configurations. For each individual patch currents were normalized to the peak current for that patch. All data points are mean ± s.e.m.

is different in these two configurations (see below). Together, these results demonstrate quantitatively that Piezo1 activation is consistent with membrane tension as the principal stimulus. In addition, this reveals that Piezo1 is a tension sensor with higher sensitivity than previously reported for other mechanically activated ion channels such as MscS and MscL, ($T_{50}$ = ~5 mN/m and ~10 mN/m, respectively, when reconstituted in asolectin liposomes or lipid bilayers (*Moe and Blount, 2005*; *Nomura et al., 2012*; *Sukharev, 2002*; *Sukharev et al., 1999*).

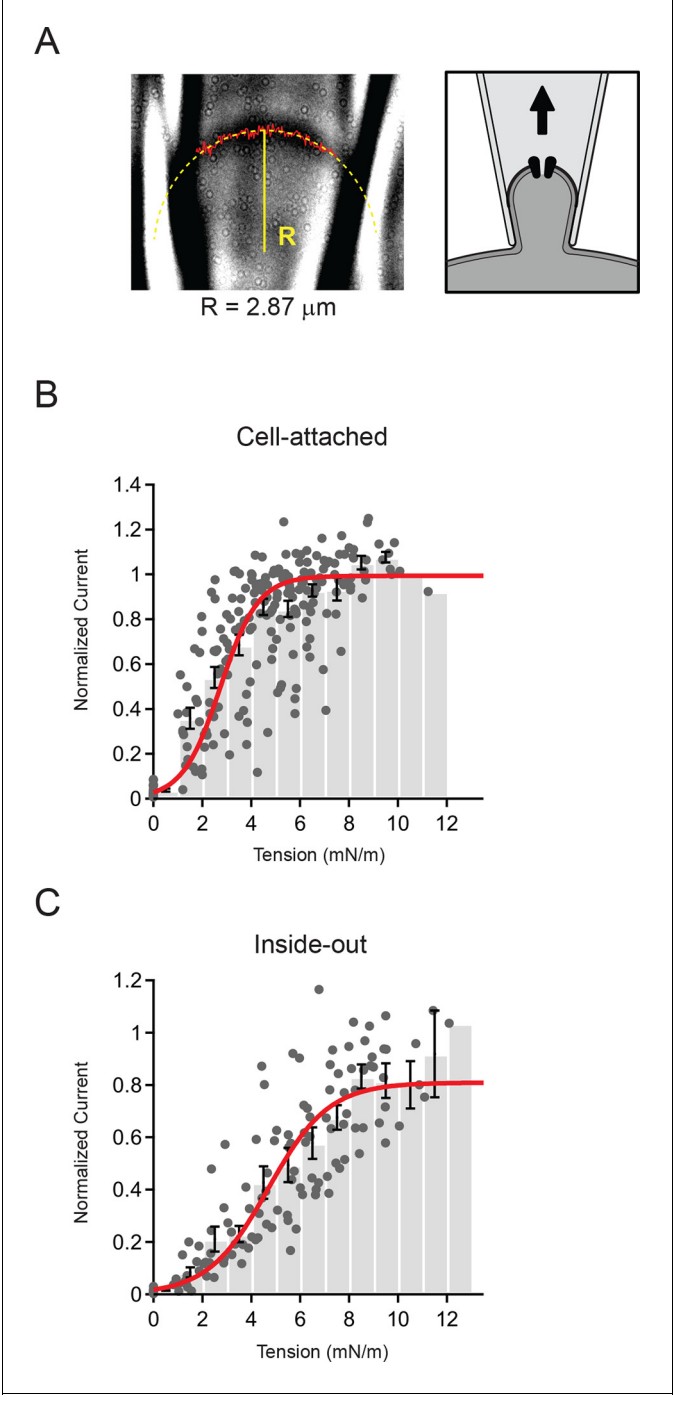

**Figure 3.** Measurement of membrane curvature and quantification of membrane tension. (**A**) Representative image of cell-attached patch and schematic showing orientation of membrane. The solid red line marks the measured position of the membrane and the dashed yellow line is a circular fit to this position. Both steps were performed using a script written in Igor Pro (see Materials and methods). For this representative patch the radius R from the fit (solid yellow line) was 2.87 µm. (**B**) Current-tension histogram for Piezo1 responses to negative pressure in cell-attached patches from HEK293t cells. For each cell, current-pressure curves were fit with a sigmoid, and each response normalized to the plateau from the fit. Tension was calculated using the measured membrane curvature from the corresponding image for each response and normalized current plotted against tension (gray circles). Data were binned (bin width 1 mN/m) and pooled (black bars; mean ± s.e.m). Binned data were fit with a Boltzmann function: $I_{max}/(1+exp(-(T-T_{50})/k]))$ where $I_{max}$ is the maximal normalized current, T is tension, $T_{50}$ is the tension of half-maximal activation, and k is the slope factor. The standard deviation of the

*Figure 3. continued on next page*

*Figure 3. Continued*

normalized amplitude for each bin was used to weight the fit. Fit parameters $I_{max}$ = 0.99±0.01, $T_{50}$ = 2.7±0.1 mN/m, k = 0.8±0.1. N = 15 cells and 218 responses. (**C**) Current-tension histogram for Piezo1 responses to negative pressure in inside-out patches from HEK293t cells. Plot was generated as described in (**B**). Fit parameters: $I_{max}$ = 0.81±0.04, $T_{50}$ = 4.7±0.3 mN/m, k = 1.2±0.1. N = 10 cells and 123 responses.

## Piezo1 responsiveness to a single pressure step is modulated by membrane tension

The resting tension in a cell-attached gigaseal patch has previously been estimated to be on the order of 0.5–4.0 mN/m, which is similar in magnitude to the tension of half-maximal activation ($T_{50}$ = 2.7 ± 0.1 mN/m) we determined in cell-attached patches (*Opsahl and Webb, 1994*). Additionally, we observed that membrane patches are already substantially curved at rest, i.e. in the absence of any external pressure difference (Δp = 0 mmHg) (*Figure 1A*). Finally, as mentioned above, we noticed in our cell-attached and inside-out recordings that stimulation with +5 mmHg was not sufficient to activate channels, but that instead a current was evoked upon pressure release (*Figures 1B and D*). We therefore hypothesized that even at rest (Δp = 0 mmHg), a substantial fraction of Piezo1 ion channels might be stimulated and subsequently inactivated. From this, we predicted that a small positive pressure stimulus, that precisely compensates the resting curvature, should effectively zero membrane tension in the patch dome and therefore, if sufficiently long, such a stimulus would allow Piezos to recover from inactivation.

To test these predictions systematically, we again utilized our ability to image and thereby precisely measure membrane curvature while simultaneously measuring Piezo1 currents. Previously, pressure prepulses have been used to modulate the resting state of $K_{2P}$ and MscS mechanosensitive ion channels prior to assaying availability (*Akitake et al., 2005*; *Honore et al., 2006*). Here, we developed a novel prepulse protocol, in which we applied pressure steps of varying amplitudes for 5 s (0—+10 mmHg, △=1 mmHg), followed by a pressure release to 0 mmHg (*Figure 4A*). Strikingly, with this first prepulse protocol we were able to elicit robust rapidly-inactivating currents in cell-attached patches not during the presence of pressure, but rather upon its release. The current amplitudes depended strongly on the prepulse amplitude in a U-shaped manner, i.e., currents were maximal after prepulses of ~ +5–6 mmHg and decreased for smaller or larger prepulses (*Figure 4A,B*). We never observed currents upon release of pressure in cells transfected with empty vector (pcDNA), indicating that these currents were indeed Piezo1-mediated (*Figure 4B*). Importantly, the biphasic dependence of current on prepulse pressure amplitude was tightly linked to membrane curvature. Specifically, we observed for each individual patch that peak currents occurred at or near the minimal curvature, i.e., when the membrane was flattest (*Figure 4C*). Averaging data from N = 14 patches further showed that currents are maximal precisely after prepulses that minimize membrane curvature (R→∞) (*Figure 4D*).

This result raised the possibility that Piezo1 ion channels could be sensing changes in tension, rather than absolute tension. However, the fact that pressure stimuli that overcompensate resting membrane tension and induce opposite curvature lead to reductions in current amplitude upon pressure release make it implausible that Piezo channels sense changes in tension. Rather, the most direct explanation is that a transient reduction in tension by flattening the membrane patch allows for recovery of Piezo1 ion channels from inactivation.

A second prediction from our hypothesis was that Piezo1 recovery from inactivation should manifest itself in a specific time course. We therefore next investigated the relationship between current amplitude and prepulse duration. First, using the above protocol, we determined for each individual patch the precise prepulse amplitude that produced the greatest current upon release of pressure. For our patch pipette sizes (typically 2–3 MΩ in our standard solutions), this was typically +5 mmHg or +6 mmHg. Second, we applied exactly this optimal prepulse stimulus for varying durations from 300 ms to 10 s (△=0.75x), followed by a return to 0 mmHg (*Figure 4E*).

Using this second prepulse protocol, we found that the increase in current amplitude with prepulse duration followed an exponential time course with τ = 2.4 ± 0.3 s (*Figure 4F,G*). This time constant is comparable to the time-dependence of recovery of Piezo2 ion channels from inactivation by whole-cell poking assays, providing further evidence that the process we are observing reflects

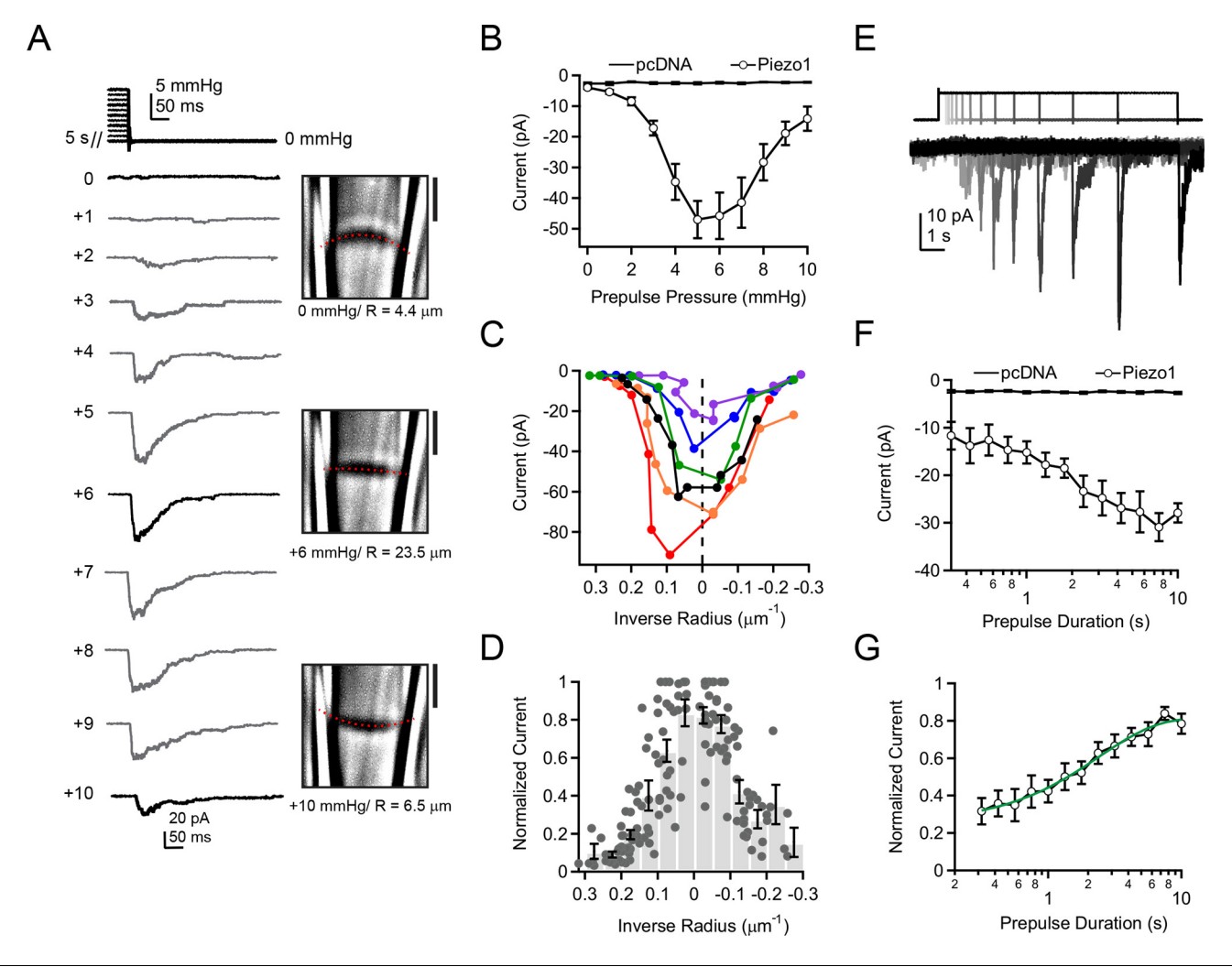

**Figure 4.** Activation of Piezo1 currents upon release of pressure stimulation. (**A**) Left, pressure stimulus protocol and representative currents showing activation of Piezo1 ion channels in a cell-attached patch upon release of a 5 s positive pressure stimulus. Holding potential was −80 mV. Right, corresponding images for 0, +6, and +10 mmHg pressure steps with membrane patch radius R fit superimposed (red dashes) and calculated radius indicated below. (**B**) Mean peak current upon release of a 5 s positive pressure pulse (0 to +10 mmHg) for cells transfected with empty vector (pcDNA; N = 9 cells) and with mouse Piezo1 (N = 14 cells). (**C**) Current-radius relationships for six representative measurements performed as shown in (**A**). The solid black line is showing the measurement in (**A**). (**D**) Normalized current-radius relationship for all measurements. For each individual patch currents were normalized to the maximal response from that patch and plotted versus inverse radius. Data were binned (bin width 0.05 µm$^{-1}$); bars represent mean normalized amplitude ± s.e.m. for each bin. N = 14 cell-attached patches and 148 responses. (**E**) Pressure-stimulus protocol and representative currents showing the time course of current increase with longer prepulse duration in a patch expressing mouse Piezo1. (**F**) Mean peak current as a function of prepulse duration for cells transfected with pcDNA or Piezo1 (N = 9 and N = 11, respectively). For each Piezo1 patch, the prepulse amplitude that caused maximal current for that cell (determined with protocol in (**A**)) was used. For our patch pipette sizes this was typically +5 or +6 mmHg; +5 mmHg was used for all pcDNA patches. (**G**) Normalized mean peak current as a function of prepulse duration for cells transfected with Piezo1. For each individual patch, currents were normalized to maximal response from that patch. Mean data were fit with an exponential function I=I$_{max}$ + A*exp(-t-t0)/tau. Fit parameters I$_{max}$ = 0.82±0.02, A = 0.49±0.02, tau = 2.4±0.3 ms. N = 11 cells. All data points are mean ± s.e.m.

recovery from inactivation (*Coste et al., 2013*). Importantly, this finding demonstrates that under standard recording conditions, a substantial fraction of Piezo ion channels is inactivated prior to pressure stimulation.

## Overall Piezo1 sensitivity is modulated by membrane tension

Thus far, our data demonstrate that membrane tension is a potent modulator of Piezo1 responsiveness to a single subsequent stimulus. With this information in hand, we next asked whether the overall sensitivity of Piezo1 could be altered by resting membrane tension. In addition to our own data, two previously reported observations suggested this might be possible. First, human gain-of-function mutations in Piezo1 that reduce channel inactivation also apparently sensitize Piezo1 to pressure stimulation (*Bae et al., 2013*). Second, the only known chemical Piezo1 agonist, Yoda1, both antagonizes inactivation and shifts the $P_{50}$ curve towards smaller values (*Syeda et al., 2015*).

To test how removing membrane tension would affect overall Piezo sensitivity, we developed a third prepulse protocol. We applied alternating 5 s prepulses of 0, +5, or +10 mmHg; each prepulse was followed by a 300 ms test pulse of varying amplitude (0 to −50 mmHg; △=5 mmHg) (*Figure 5A*). We chose these precise prepulse amplitudes because we had previously observed that +5 mmHg prepulses nearly flattened the membrane, while +10 mmHg prepulses induced opposite (concave) curvature that was roughly equivalent in magnitude to the resting (convex) curvature (*Figure 4A*). We chose a 5 s prepulse duration because this was sufficient for nearly complete recovery from inactivation while minimizing premature rupture of patches during long pulses to positive pressures (*Figure 4E*). As before, this experiment was performed with simultaneous imaging of cell-attached membrane patches.

The effect of the +5 mmHg prepulse on the overall current-pressure relationship was striking: When preceded by a +5 mmHg prepulse, currents elicited by a subsequent test pulse were greatly increased in amplitude. Additionally, the pressure of half-maximal activation was shifted by ~9 mmHg towards lower pressures ($P_{50}$ = −16.8 ± 2.8 mmHg with a 0 mmHg prepulse and $P_{50}$ = −7.7 ± 1.1 mmHg with a +5 mmHg prepulse; N=11; P<0.05) (*Figure 5A and B*). Importantly, prepulses of +10 mmHg did not affect the overall pressure sensitivity as compared to 0 mmHg prepulses ($P_{50}$ = −16.8 ± 2.8 mmHg with a 0 mmHg prepulse and $P_{50}$ = −13.5 ± 2.9 mmHg with a +10 mmHg prepulse; N=11; P>0.05). As before (*Figure 2A*), only negligible currents were elicited in patches transfected with empty vector (pcDNA), even during test pulses preceded by a +5 mmHg prepulse, indicating the increase in current at lower pressures did not result from novel recruitment of endogenous mechanosensitive channels in HEK293t cells (*Figure 5B*). These results suggest that pressure prepulses that minimize membrane tension shift overall Piezo sensitivity maximally leftwards.

Importantly, we found that during the long duration of the prepulse experiment, membrane geometry was sufficiently stable to reversibly and reliably alternate between different membrane curvatures (*Figure 5C*). While there are slight variations in resting radius throughout the duration of the recording, likely due to slight creep of the patch, these are minor compared to the large changes in radius induced by the pre- and test pulses.

To establish the effect of prepulses on overall Piezo sensitivity quantitatively, we calculated membrane tension during the test pulse from the corresponding images. We found that the tension of half-maximal activation was indeed leftward shifted, from $T_{50}$ = 2.2 ± 0.1 mN/m without a prepulse (0 mmHg) to $T_{50}$ = 1.4 ± 0.1 mN/m with a +5 mmHg prepulse (P<0.001; N=11; *Figure 5D,E*, and G). Upon a +10 mmHg prepulse, the $T_{50}$ was also leftward shifted to that without a prepulse, but to a lesser extent than with a +5 mmHg prepulse ($T_{50}$ = 1.8 ± 0.2 mN/m; P=0.03; *Figure 5F and G*). The slope factor was unaltered by prepulse amplitude (0 mmHg prepulse, k = 0.8 ± 0.1; +5 mmHg prepulse, k = 0.7 ± 0.1; P=0.53 vs 0 mmHg; +10 mmHg prepulse, k = 1.1 ± 0.2; P=0.07 vs. 0 mmHg). Altogether, our approach unmasks the inherent tension sensitivity of Piezo1 and demonstrates that it can be substantially modulated by resting membrane tension. Our results imply that in a native and undisturbed cell, the same mechanism might affect overall sensitivity of Piezo1.

## Discussion

We originally set out to identify the physical stimulus that activates Piezo1 ion channels. We chose to examine Piezo sensitivity in membrane patches from living cells, which differs from the bottom-up approaches of artificial lipid bilayers or micelle systems, in that the lipid composition is heterogeneous, the channel protein is not purified from possible interaction partners, and cellular mechanical stability is maintained. We found that similar to the vertebrate mechanosensitive $K^+$ channels TREK-1 and TRAAK, Piezo1 responds robustly to both positive and negative pressure (*Brohawn et al., 2014b*). There are several possible mechanisms for how mechanosensitive ion channels may convert

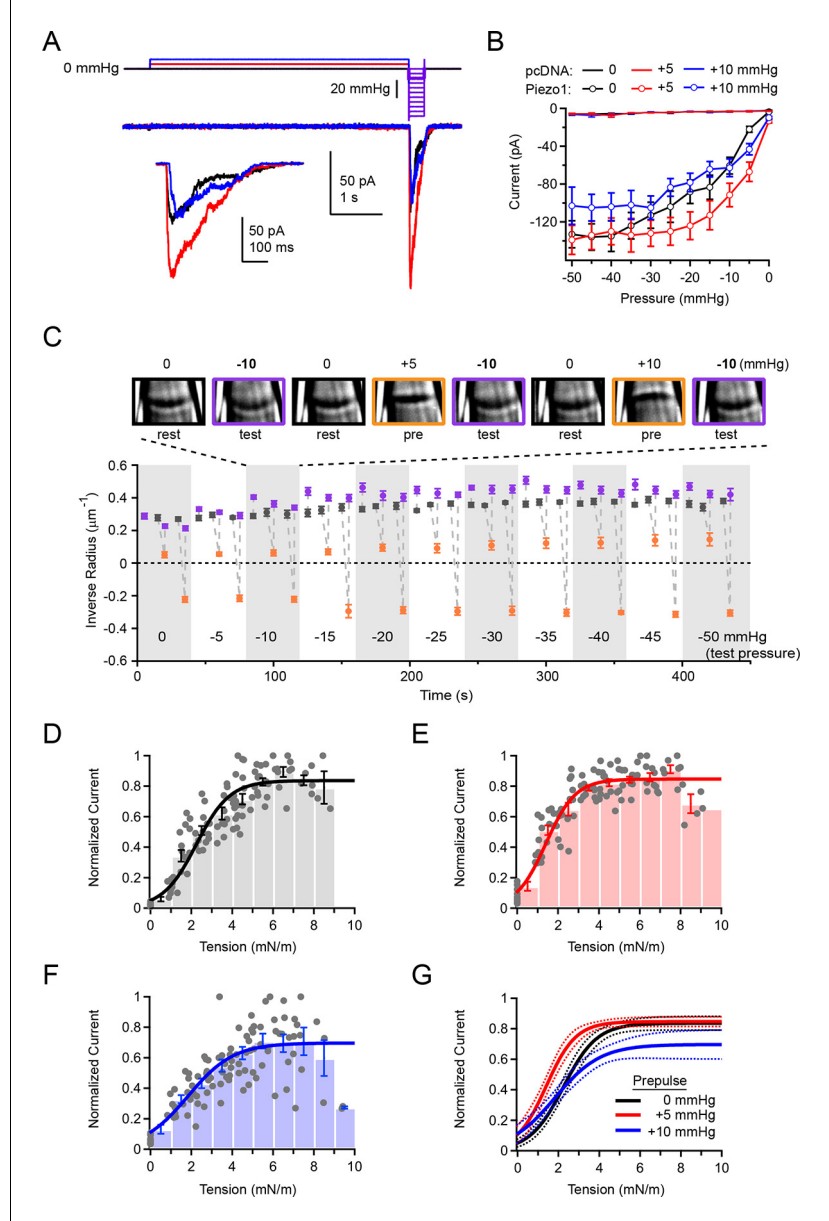

**Figure 5.** Overall Piezo1 sensitivity is regulated by resting membrane tension. (**A**) Stimulus protocol and representative currents from a cell-attached HEK293t cell patch expressing mouse Piezo1-IRES-GFP. The test pulse for these currents was −10 mmHg (thick purple line); holding potential was −80 mV. Inset shows test currents at magnified scale. (**B**) Peak current-pressure relationships for test pulses (0 to −50mmHg, Δ5 mmHg) following 5 s 0 mmHg, +5 mmHg and +10 mmHg prepulses.. All data points are mean ± s.e.m. N = 8 cell-attached patches (pcDNA) and 11 cell-attached patches (Piezo1). (**C**) Mean patch curvature as a function of time during protocol performed shown in (**A**). Representative images of one individual patch are shown above. Each patch was tested with no prepulse (0 mmHg), a +5 mmHg prepulse, and a +10 mmHg prepulse at each test pressure before advancing to the next test pressure. Grey markers show inverse radius during rest periods (0 mmHg, between stimuli), purple markers show inverse radius during 300 ms test pulses (0 to −50 mmHg, Δ5 mmHg), orange markers show inverse radius during +5 mmHg or +10 mmHg prepulse. All data points are mean ± s.e.m. N = 11 for cell-attached patches. (**D–F**) Normalized current-tension relationships obtained from protocol shown in (**A**) using no prepulse (0 mmHg) (**D**), +5 mmHg prepulse (**E**) and +10 mmHg prepulse (**F**). Currents from individual patches are normalized to the maximal response for each patch. Data were pooled and binned (bin width 1 mN/m); bars represent mean ± s.e.m. N = 11 patches. Binned data were fit with a Boltzmann function $I = I_{max}/(1+exp(-(T-T_{50}/k))$ where I is normalized current, $I_{max}$ is the plateau, T is tension, $T_{50}$ is the tension of half-maximal activation, and k is the slope factor. The standard deviation of the normalized amplitude for each bin was used to

*Figure 5. continued on next page*

*Figure 5. Continued*

weight the fit. Fit parameters for no prepulse (0 mmHg): $I_{max}$ = 0.84±0.02, $T_{50}$ = 2.2±0.1 mN/m, k = 0.8±0.1. For +5 mmHg prepulse: $I_{max}$ = 0.85±0.01, T--$_{50}$ = 1.4±0.1 mN/m, k = 0.7±0.1. For +10 mmHg prepulse: $I_{max}$ = 0.70±0.04, $T_{50}$ = 1.8±0.2 mN/m, k = 1.1±0.2. (G) Fits from D-F overlayed (solid line) with 95% confidence intervals (dashed lines).

physical force into pore opening (*Ranade et al., 2015*; *Sukharev and Corey, 2004*). In a tethered mechanism, force could be transmitted to the channel through auxiliary proteins, whereas in a bilayer mechanism, force could be transmitted directly through the lipid bilayer. In the latter case, the channel must sense either membrane curvature or lateral membrane tension. Several well-studied mechanosensitive ion channels have previously been demonstrated to sense lateral membrane tension, including prokaryotic channels MscS and MscL, as well as TREK-1 and TRAAK (*Brohawn et al., 2014b*; *Moe and Blount, 2005*; *Sokabe et al., 1991*; *Sukharev, 1999*; *2002*). However, Piezos are distinct from each of these previously identified tension sensors in that they are much larger, with many more predicted transmembrane domains, and in that they share no homology on a primary sequence or overall architectural level (*Brohawn et al., 2014a*; *Coste et al., 2015*; *Ge et al., 2015*; *Kamajaya et al., 2014*).

Our simultaneous imaging with electrophysiology revealed that both convex and concave macroscopic curvature in the membrane patch induce channel opening. While it is theoretically possible that Piezo1 senses convex and concave membrane curvature with equal sensitivity, this mechanism would require curvature sensing structures that are symmetrical. However, Piezo proteins do not contain any amino acid sequences with similarity to any known curvature-sensing proteins, and the recently obtained cryo-electron microscopy structure shows no symmetrical features with respect to the plane of the bilayer, making this possibility unlikely (*Antonny, 2011*; *Ge et al., 2015*; *McMahon and Boucrot, 2015*). It is also possible that local membrane curvature and tension are not strictly coupled to macroscopic membrane curvature. Previous reports indicate that alterations in local curvature induced by asymmetric incorporation of lipids can change the response of tension-gated channels to pressure (*Perozo et al., 2002*); this will also have to be tested for Piezo1.

Consistent with the idea that Piezo1 senses lateral membrane tension, we were able to combine electrophysiology and imaging to quantify the tension required for activation of Piezo1 in cell-attached and inside-out patches. Our data suggest that Piezo is a unique mechanosensor, as Piezo1 probed in a cellular environment is much more sensitive to tension ($T_{50}$ = 1.4 ± 0.1 mN/m) than are MscL and MscS in azolectin liposomes ($T_{50}$ ~5–10 mN/m), although we cannot rule out differences arising from the different local lipid environment in eukaryotic cells versus reconstituted systems (*Moe and Blount, 2005*; *Nomura et al., 2012*; *Sukharev, 2002*; *Sukharev et al., 1999*). Together, our data support the evolutionary need for this novel mechanosensor; unlike the prokaryotic 'release valves' MscL and MscS, Piezo is able to detect very small changes in membrane tension, an ability that is essential for its role in intricate physiological processes ranging from sensing renal and blood flow to detection of light touch (*Cahalan et al., 2015*; *Maksimovic et al., 2014*; *Ranade et al., 2014a*; *Ranade et al., 2014b*; *Woo et al., 2014*).

Observing Piezo1 activity in multiple patch configurations also allowed us to make several observations. First, sensitivity of Piezo1 to tension differed in cell-attached versus inside-out patches. One potential explanation for this difference is the varying amount of cytoskeleton retained in the two configurations, which contributes to membrane properties, including tension (*Gauthier et al., 2012*). Consistent with this, we observed lower optical density in inside-out patches, which are known to contain less cytoskeleton (*Suchyna et al., 2009*). Together, this predicts that the cytoskeleton is an important regulator of Piezo1 sensitivity, a finding that is reconciled with previous reports that inhibition of actin polymerization with cytochalsin-D inhibits whole-cell Piezo1 currents evoked by direct stimulus with a glass pipette, but increases opening in cell-attached pressure-evoked currents (*Gnanasambandam et al., 2015*; *Gottlieb et al., 2012*). In fact, we cannot rule out a role of the cytoskeleton in directly contributing or even being essential to Piezo activation. We also observed differences in both rate and extent of decay of Piezo1-mediated currents among patch configurations. While the mechanism of Piezo1 inactivation is unknown, a previous study reports that inactivation is irreversibly lost with repeated stimulation, which may explain why we observed the least

inactivation in outside-out patches, in which the membrane undergoes the most manipulation prior to assaying activity (*Gottlieb et al., 2012*).

Second, the reported sensitivity of Piezo1 to pressure shows remarkable variation among cell types and stimulation protocols. For example, while the $P_{50}$ is typically reported to be ~ −30 mmHg for heterologously expressed Piezo1 in HEK293t cells, in neural stem cells, the $P_{50}$ was ~−13 mmHg, whereas the $P_{50}$ was ~−40 mmHg in the breast cancer cell line MCF-7 (*Coste et al., 2010*; *Li et al., 2014*; *Pathak et al., 2014*). One explanation for this is that Piezo1 sensitivity is modulated by interaction with other cellular components, which may be differentially expressed: For example, Piezo1 mechanosensitivity requires the presence of phosphoinositides, which are depleted over time in excised patches (*Borbiro et al., 2015*); the integral membrane protein STOML3 also greatly increases Piezo1 sensitivity (*Poole et al., 2014*). Here, we found that an additional modulator of Piezo sensitivity is resting membrane tension. Importantly, cellular membrane tension varies greatly with cell type; even within one cell local tension depends on factors including lipid composition, cytoskeletal contacts, the extracellular matrix, and others (*Blumenthal et al., 2014*; *Hoffman, 2014*; *Vasquez et al., 2014*). As Piezos are thus under differing tension depending on their cellular expression and localization, the ability of tension to modulate sensitivity gives Piezo1 a broad tuning curve that primes it to respond to physiologically relevant changes in tension at that location. Importantly, this makes Piezos robust sensors of membrane tension in the remarkably wide variety of cell types in which they are expressed.

Third, our results also identify inactivation as an important physiological modulator of overall Piezo1 sensitivity. Interestingly, the currents we observed after Piezo channels recovered from inactivation during a transient period of zero tension (i.e., +5 mmHg prepulses) may have important physiological relevance: a stimulus that results in a local reduction of membrane tension may lead to increased Piezo activity in this region upon release of this stimulus, perhaps providing a mechanism by which cells can sense not only the onset, but also the offset of a stimulus. The interplay between activation and inactivation may also make Piezo1 most sensitive to rapidly applied mechanical stimuli, similar to previous reports for MscS, as slowly applied stimuli will lead to gradual accumulation of channels in inactivated states (*Akitake et al., 2005*). The advent of structural information about the various domains of Piezo, as well as its overall architecture will be extremely useful in identifying the structural correlates of both the mechanosensor and the inactivation mechanism, as activation and inactivation combine to dictate overall sensitivity (*Ge et al., 2015*; *Kamajaya et al., 2014*).

Finally, we anticipate that our simple prepulse protocol will provide a useful tool for measuring the inherent mechanosensitivity in different cells and irrespective of inactivation kinetics by manipulating the curvature of the membrane to minimize tension prior to testing sensitivity. The prepulse amplitude required for flattening of the membrane and removal of resting tension will vary with the particulars of a given system, including cell type and pipette size, but can be measured even in the absence of imaging data with the protocol in *Figure 4B*, by using the pressure at which the maximal offset current amplitude is evoked.

## Materials and methods

### Cell culture

Human embryonic kidney HEK293t cells (ATCC # 3579061) were provided and authenticated (STR authenticated and verified mycoplasma-free) by the Duke Cell Culture Facility. Cells were grown in DMEM (Life Technologies) with 10% heat-inactivated fetal bovine serum (Clontech Laboratories, Mountain View, CA), 50 units/ml penicillin, and 50 mg/ml streptomycin (Life Technologies, Carlsbad, CA). Cells were transiently transfected in 6 well plates in the presence of 10 µM ruthenium red with Mouse Piezo1-IRES-GFP (3 µg) or empty vector (pcDNA3.1(-) and GFP) using Fugene (Promega, Madison, WI) ~48 hr before recording. Transfected cells were reseeded at low density the day before recording in 50 mm glass-bottomed dishes (P50G-0-30-F; MatTek Corporation, Ashland, MA) coated with Poly-L-lysine and laminin.

### Electrophysiology

Patch-clamp recordings were performed at room temperature using an EPC10 amplifier and Patchmaster software (HEKA Elektronik, Lambrecht, Germany). Data were sampled at 5 kHz and filtered

at 2.9 kHz. Borosilicate glass pipettes (1.5 OD, 0.85 ID; Sutter Instrument Company, Novato, CA) had a resistance of 1.5–4 MΩ when filled with pipette buffer solution (in mM: 130 NaCl, 5 KCl, 10 HEPES, 10 TEACl, 1 CaCl$_2$, 1 MgCl$_2$, pH = 7.3 with NaOH). The standard bath solution was (in mM): 140 KCl, 10 HEPES, 1 MgCl$_2$, 10 glucose, pH = 7.3 with KOH. Pipettes were angled at ~15° with respect to the glass cover slip to optimize image contrast. Pressure was controlled with a high-speed pressure clamp system (HSPC-1; ALA Scientific Instruments, Farmingdale, NY). Patches were held at −80 mV and stimulated with pressure-step protocols described in the manuscript. Unless stated otherwise, sweeps were separated by 10 s to allow for recovery from inactivation.

Electrophysiological recordings were only analyzed for patches with a seal resistance of at least 1GΩ and maximal pressure-induced currents of at least 50 pA for cell-attached and inside-out patches with negative pressure and 20 pA for all other configurations. Only one patch was excised from each cell; with the exception of *Figure 4F*, which required prior determination of the appropriate prepulse for each patch using the protocol in 4B, only one protocol was performed on each patch. Analysis was performed with Igor Pro 6.22A (WaveMetrics, Lake Oswego, OR). Baseline currents before pressure stimulation were subtracted off-line and peak currents measured at each pressure. Student's t-test or ANOVA followed by Tukey-Kramer comparison of pairs of means were used to assess statistical significance.

### Imaging

Images of the cell membrane inside the patch pipette were captured at a rate of ~7.5 frames per second (125 ms exposure) at a resolution of 61.5 pixels/μm using a Plan Apo (100x) DIC oil objective coupled with a Coolsnap ES camera and 4x relay lens (Nikon Instruments Inc, Melville, NY). During imaging, the focal plane was continuously adjusted to center on the contact points of the membrane with pipette walls, indicated by the 'crossing over' of the lines corresponding to the pipette (see *Video 1*). Short (300 ms) pressure pulses were used to minimize membrane 'creep' and excessive movement of the membrane out of the focal plane. Images were extracted from videos in NIS-Elements (Nikon Instruments Inc, Melville, NY) and imported into Igor Pro (WaveMetrics, Lake Oswego, OR). To identify the membrane geometry, a line scan parallel to the pipette walls was performed to localize the minimum pixel intensity for each line over a rolling average of 9 pixels; the script, custom-written in IgorPro, is available in our Github Repository (github.com/GrandlLab). These positions were then fit with a circle to obtain the radius (R). Tension (T) was calculated for every pressure step $\Delta p$ using Laplace's law: $T = R \cdot \Delta p / 2$. For pooling and binning data, current-pressure responses for individual cells were fit with a Boltzmann function ($I_{max}/(1+\exp(-(P-P_{50})/k))$) and individual current amplitudes were re-normalized to the plateau response to saturating stimuli from the fit ($I_{Max}$). To calculate the tension required for activation, binned data were fit with a Boltzmann function ($I = I_{max}/(1+\exp(-(T-T_{50})/k))$); the fit was weighted with the standard deviations from each bin.

## Acknowledgements

This study was supported by the Klingenstein Fund and Duke University. We thank Dr. Bertrand Coste, Ardem Patapoutian and Geoff Pitt, and all members of the Grandl lab for thoughtful comments on study and manuscript and Jason Wu for support with figure design.

## Additional information

### Funding

| Funder | Author |
| --- | --- |
| Duke University School of Medicine | Amanda H Lewis<br>Jörg Grandl |

The funders had no role in study design, data collection and interpretation, or the decision to submit the work for publication.

## Author contributions

AHL, Conception and design, Acquisition of data, Analysis and interpretation of data, Drafting or revising the article; JG, Conception and design, Analysis and interpretation of data, Drafting or revising the article

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
