## [Decision Letter]

Thank you for submitting your work entitled "Mechanical sensitivity of Piezo1 ion channels can be tuned by cellular membrane tension" for consideration by *eLife*. Your article has been reviewed by three peer reviewers, and the evaluation has been overseen by Richard Aldrich as the Reviewing Editor and Senior Editor.

The reviewers have discussed the reviews with one another and the Reviewing editor has drafted this decision to help you prepare a revised submission.

Summary:

In this manuscript, the authors provide an in-depth assessment of the nature of the mechanical stimulus that activates Piezo1 channels in heterologous cells. In brief, there are two key findings. First, Piezo1 channels respond to changes in membrane tension rather than curvature. Second, that sustained tensions even such as those present in membrane patches held at symmetric atmospheric pressures are sufficient to inactivate a proportion of the channels. This leads to activation following applications of pressures sufficient to flatten patches and may help to establish resting set points in cells. These biophysical findings were facilitated by careful measurements of membrane curvature as well as controlled application of pressure differences across membrane patches. They have important applications for how Piezo1 works in cells and for the origin of variations in the kinetics of their responses to mechanical stimulation.

Essential revisions:

1) The authors argue that the cytoskeleton is unlikely to contribute to Piezo1 channel activation on the basis of the similarity of responses observed in cell-attached and excised patch conditions. While it is reasonable to assume that the cytoskeleton is disrupted in excised patches, direct examination of this question has indicated that excised patches can retain their association with the cortical cytoskeleton. The authors should either perform additional experiments to directly address (e.g. determine how application of cytoskeleton disrupting agents like cytocholasinD affect tension-sensitivity) this inference or rephrase the discussion of this result-indicating that while their data are more consistent with the idea that the cytoskeleton is dispensable for Piezo1 activation in HEK293T cells, that they cannot exclude this possibility

2) The most novel contribution of this work is the measurement of membrane tension/curvature while recording from piezo1-containing patches. This is challenging, and perhaps a more thorough description of the imaging procedure would be helpful. For instance, it is unclear how the authors ensure that they are looking at a plane that crosses the center of the membrane patch. If this is not the case, curvature estimates might be quite inaccurate. Also, the custom-script written in Igor-Pro to measure the radius of membrane curvature should be made available, at least upon request, if not through a free source code repository (GitHub or the like).

3) The authors use a clever pre-pulse protocol to "wake up" inactivated piezo1 channels. Use of pre-pulses is not new in the study of mechanosensitive channels and it has been used to study inactivation of MscS (Akitake et al., JGP 2005) and to show that "the availability of MscS depends on both the rate of pressure application and the amplitude of prepulse pressure". It would be informative for the readers to discuss the piezo1 results in the context of prior work with pre-pulses and mechanosensitive channels.

4) The authors show control recordings obtained when using an empty vector for transfection, which is quite good. The recorded currents are mostly flat, but there seems to be some minuscule response at small positive pressures in inside-out patches (Figure 2). This might be within the noise, but scales are all different in panels A-C, so it is difficult to appreciate the relevance of the small currents, and I do note that data presented in Figure 4 is right on the same range of pressures and currents of the signal seen for pcDNA transfected cells (Figure 2). In addition, no controls are presented for experiments with pre-pulses, and there is the possibility (admittedly very minor) that the authors are "waking up" other channels present in HEK293t cells. Since pre-pulses are right on the range of 5 to 10 mmHg, it would be important to see a control curve with vector-only transfected cells for pre-pulse protocols. Also, please indicate if the data presented in Figure 4 and Figure 5 were obtained using inside-out patches.

5. One important point to clarify is that the authors claim that the curvature (positive or negative) of the membrane is not important for Piezo1 activation, thus concluding that membrane tension is the most important factor in activation. However, it is not clear that the local curvature around the channels (which changes in a scale of a few nanometers) can be affected by the macroscopic patch curvature (happening in a micrometer scale). It has been seen in gramicidin channels that altering the local curvature by asymmetric lipids can change the coupling to lateral tension.

---

## [Author Response]

*Essential revisions: 1) The authors argue that the cytoskeleton is unlikely to contribute to Piezo1 channel activation on the basis of the similarity of responses observed in cell-attached and excised patch conditions. While it is reasonable to assume that the cytoskeleton is disrupted in excised patches, direct examination of this question has indicated that excised patches can retain their association with the cortical cytoskeleton. The authors should either perform additional experiments to directly address (e.g. determine how application of cytoskeleton disrupting agents like cytocholasinD affect tension-sensitivity) this inference or rephrase the discussion of this result-indicating that while their data are more consistent with the idea that the cytoskeleton is dispensable for Piezo1 activation in HEK293T cells, that they cannot exclude this possibility*

This is an excellent point and by no means did we intend to convey the impression that we conclude from our data that the cytoskeleton is dispensable for Piezo activation. In our own defense, in the original manuscript we already wrote: “Moreover, this predicts that the cytoskeleton is an important regulator of Piezo sensitivity, …”.

We considered the design of a thorough experiment to test this question. However, we face the issue that cytoskeletal disrupting agents (for example, cytoD or latrunculin B) are not acute disrupters that depolymerize actin, but instead inhibit polymerization, and thus act slowly compared to the lifetime of a patch. This limitation is also supported by a publication (Gottlieb, 2012) that reports that chronic treatment with Cyto-D does not affect sensitivity of Piezo1 in the cell-attached mode. The best way to test whether the cytoskeleton is dispensable for Piezo1 activation would be to reconstitute the protein in bilayers and show that it retains mechanosensitivity, which is beyond the scope of this study. We therefore decided not to perform such experiments.

To drive home this important point we added a sentence to our manuscript: “In fact, we cannot rule out a role of the cytoskeleton in directly contributing or even being essential to Piezo activation.”

*2) The most novel contribution of this work is the measurement of membrane tension/curvature while recording from piezo1-containing patches. This is challenging, and perhaps a more thorough description of the imaging procedure would be helpful. For instance, it is unclear how the authors ensure that they are looking at a plane that crosses the center of the membrane patch. If this is not the case, curvature estimates might be quite inaccurate. Also, the custom-script written in Igor-Pro to measure the radius of membrane curvature should be made available, at least upon request, if not through a free source code repository (GitHub or the like).*

We added several additional sentences to the Materials and methods section describing the imaging protocol in greater detail: “During imaging, the focal plane was continuously adjusted to center on the contact points of the membrane with pipette walls, indicated by the “crossing over” of the lines corresponding to the pipette. Short (300 ms) pressure pulses were used to minimize membrane “creep” and excessive movement of the membrane out of the focal plane.”

We also specified the manufacturer of the dishes we used: “P50G-0-30-F; MatTek Corporation, Ashland, MA”.

To better help other labs reproducing these experiments we now also provide a video file of a representative experiment Video 1.

Finally, we will of course happily provide the custom code in IgorPro (Wavemetrics) used to find the membrane. Specifically, we uploaded the code to a Github repository and added a note to the Materials and methods section:”the script, custom-written in IgorPro, is available in our Github Repository (github.com/GrandlLab).

*3) The authors use a clever pre-pulse protocol to "wake up" inactivated piezo1 channels. Use of pre-pulses is not new in the study of mechanosensitive channels and it has been used to study inactivation of MscS (Akitake et al., JGP 2005) and to show that "the availability of MscS depends on both the rate of pressure application and the amplitude of prepulse pressure". It would be informative for the readers to discuss the piezo1 results in the context of prior work with pre-pulses and mechanosensitive channels.*

This is an excellent suggestion and we have added this proposed reference (Akitake et al., JGP 2005) and one additional reference (Honore, et al., PNAS 2006) to the following statements in the Results section:”Previously, pressure prepulses have been used to modulate the resting state of K_2P_ and MscS mechanosensitive ion channels prior to assaying availability” and the Discussion: “The interplay between activation and inactivation may also make Piezo1 most sensitive to rapidly applied mechanical stimuli, similar to previous reports for MscS, as slowly applied stimuli will lead to gradual accumulation of channels in inactivated states”, respectively.

*4) The authors show control recordings obtained when using an empty vector for transfection, which is quite good. The recorded currents are mostly flat, but there seems to be some minuscule response at small positive pressures in inside-out patches (Figure 2). This might be within the noise, but scales are all different in panels A-C, so it is difficult to appreciate the relevance of the small currents, and I do note that data presented in Figure 4 is right on the same range of pressures and currents of the signal seen for pcDNA transfected cells (Figure 2). In addition, no controls are presented for experiments with pre-pulses, and there is the possibility (admittedly very minor) that the authors are "waking up" other channels present in HEK293t cells. Since pre-pulses are right on the range of 5 to 10 mmHg, it would be important to see a control curve with vector-only transfected cells for pre-pulse protocols. Also, please indicate if the data presented in Figure 4 and Figure 5 were obtained using inside-out patches.*

This is an excellent point, and we have repeated prepulse experiments with cells transfected with empty vector (pcDNA). In 9 patches, we never observed currents upon release of a pre-pulse (i.e., +5 mmHg to 0 mmHg). We also repeated P50 protocols with prepulses in 8 pcDNA-transfected cells. While we again observed occasional small (<10 pA) currents in non-transfected cells, the magnitude of these currents was unaffected by prepulse amplitude, and thus they do not contribute to the primary effect we observed. We analyzed these data and added them to the manuscript in Figure 4 and Figure 5 and corresponding text: “We never observed currents upon release of pressure in cells transfected with empty vector (pcDNA), indicating that these currents were indeed Piezo1-mediated (Figure 4)” and: “As before (Figure 2), only negligible currents were elicited in patches transfected with empty vector (pcDNA), even during test pulses preceded by a +5 mmHg prepulse, indicating the increase in current at lower pressures did not result from novel recruitment of endogenous mechanosensitive channels in HEK293t cells (Figure 5)”.

We have also modified Figure 5 to display mean raw, rather than normalized currents. Also, the data in Figure 4 and Figure 5 are all obtained from cell-attached patches, and this is now stated in the manuscript text:”As before, this experiment was performed with simultaneous imaging of cell-attached membrane patches.”

*5. One important point to clarify is that the authors claim that the curvature (positive or negative) of the membrane is not important for Piezo1 activation, thus concluding that membrane tension is the most important factor in activation. However, it is not clear that the local curvature around the channels (which changes in a scale of a few nanometers) can be affected by the macroscopic patch curvature (happening in a micrometer scale). It has been seen in gramicidin channels that altering the local curvature by asymmetric lipids can change the coupling to lateral tension.*

This is an interesting possibility, and we have clarified our claim in the Discussion to indicate that our experiments show that while Piezo1 activity responds to global curvature in either direction, local curvature may differ and could still play a role in gating:”It is also possible that local membrane curvature is not strictly coupled to macroscopic membrane curvature. Previous reports indicate that alterations in local curvature induced by asymmetric incorporation of lipids can change the response of tension-gated channels to pressure (Perozo et al., 2002); this will also have to be tested for Piezo1.”